# Learning as filtering: Implications for spike-based plasticity

**Jannes Jegminat**[1,2]*, **Simone Carlo Surace**[1], **Jean-Pascal Pfister**[1,2]

**1** Department of Physiology, University of Bern, Bern, Switzerland, **2** Institute of Neuroinformatics and Neuroscience Center Zurich, ETH and the University of Zurich, Zurich, Switzerland

* jannes@ini.uzh.ch

## Abstract

Most normative models in computational neuroscience describe the task of learning as the optimisation of a cost function with respect to a set of parameters. However, learning as optimisation fails to account for a time-varying environment during the learning process and the resulting point estimate in parameter space does not account for uncertainty. Here, we frame learning as filtering, i.e., a principled method for including time and parameter uncertainty. We derive the filtering-based learning rule for a spiking neuronal network—the Synaptic Filter—and show its computational and biological relevance. For the computational relevance, we show that filtering improves the weight estimation performance compared to a gradient learning rule with optimal learning rate. The dynamics of the mean of the Synaptic Filter is consistent with spike-timing dependent plasticity (STDP) while the dynamics of the variance makes novel predictions regarding spike-timing dependent changes of EPSP variability. Moreover, the Synaptic Filter explains experimentally observed negative correlations between homo- and heterosynaptic plasticity.

## Author summary

The task of learning is commonly framed as parameter optimisation. Here, we adopt the framework of learning as filtering where the task is to continuously estimate the uncertainty about the parameters to be learned. We apply this framework to synaptic plasticity in a spiking neuronal network. Filtering includes a time-varying environment and parameter uncertainty on the level of the learning task. We show that learning as filtering can qualitatively explain two biological experiments on synaptic plasticity that cannot be explained by learning as optimisation. Moreover, we make a new prediction and improve performance with respect to a gradient learning rule. Thus, learning as filtering is a promising candidate for learning models.

## 1 Introduction

In computational neuroscience, most normative models frame learning as optimisation of a static cost function with respect to a set of parameters, such as synaptic efficacies [1–9] or a

---

**Data Availability Statement:** All software and data files are available from github (doi.org/10.5281/zenodo.3970146) and https://github.com/Theoretical-Neuroscience-Group/SynapticFilter.jl.

**Funding:** This research was supported by the Swiss National Science Foundation grants

PP00P3_179060 (JJ, SCS, JPP) and 31003A_175644 (JJ, SCS, JPP), and by the Institute of Physiology in Bern (JJ, SCS, JPP). The funders had no role in study design, data collection and analysis, decision to publish, or preparation of the manuscript.

**Competing interests:** The authors have declared that no competing interests exist.

neuron's excitability [10, 11]. Different cost functions have been used to reproduce or predict experimental findings, such as a measure of sparseness and information preservation [12], mutual information [1], the probability of timed postsynaptic spiking [13, 14], the mutual information of input and output spike trains [15, 16], the network sensitivity [17] and free energy [18].

However, learning as optimisation has the drawback of not taking parameter uncertainty into account [19]. When few training data are available compared to the number of model parameters, the parameter space is not sufficiently constrained, i.e., multiple parameter instantiations yield comparable model performance. Optimisation selects the best performing parameter, thereby ignoring the inherent parameter uncertainty present in a (probabilistic) model. This contributes to the problem of overfitting, i.e., the resulting performance on the training data does not generalise to the testing data [20, 21] Moreover, many decision-making models require as input not only the most likely prediction but also prediction uncertainty [22]. To obtain accurate prediction uncertainty, the contribution of parameter uncertainty must be taken into account (e.g. [23]). Thus parameter uncertainty is computationally relevant for avoiding overfitting and the estimation of prediction uncertainty.

Learning as static optimisation is further limited because it lacks a principled way of accounting for a dynamic environment during learning. Often the data distribution is assumed to be static, i.e., independent of time. However, in many settings the environment and, thus, the data distribution, are dynamic. For example, the association between a location and the availability of food is not static when the source of food depletes or moves over time. For the learner, dynamic environments pose the additional challenge of determining the speed of learning. A slow learner fails to adapt to quickly changing environmental statistics while an overly fast learner might disregard past data prematurely. The question of how to account for a dynamic environment during learning is closely related to continual learning, i.e., the task of sequentially learning from multiple data sets while maintaining (testing) performance on all previously observed ones [24, 25]. Here, the dynamics of the environment translate into the sequential availability of data sets.

The relation between time and uncertainty has been addressed in neuroscience under several conditions [26, 27]. For instance, flies learn odour association and adapt their forgetting rate of old associations [28, 29]. Similar experiments have been conducted with rodents [30, 31]. Uncertainty of rewards has been studied in the prefrontal and cingulate cortex based on reinforcement learning models [32]; and several neuro-modulators have been identified that influence choices under uncertainty [33, 34]. The uncertainty related to a whisker stimulus is directly related to neuronal activity in the rat barrel cortex [35]. Uncertainty has also been linked to neuronal codes [35, 36] and many aspects of perception and decision making [37]. In the context of plasticity, uncertainty of synaptic weights has been linked to spine turnover [38].

Normative models of learning can benefit from going beyond the framework of static optimisation by including parameter uncertainty and time in the learning task. However, it remains unclear which framework could prove to be a fruitful alternative. Here, we propose to address the problem of learning by using the filtering framework. Filtering is the preferential way to include time and uncertainty since it continuously computes the posterior distribution (also called the filtering distribution) of a latent variable from all the observations up to time $t$. Filtering theory was first developed for linear problems [39–42] and then generalised to a proper nonlinear filtering by mathematicians in the 60's [43, 44]. For a review on nonlinear filtering, see [45, 46]). See also [39] for practical applications of linearised filters. We apply learning as filtering to synaptic plasticity, a field in which the need for new learning paradigms has become apparent [47].

In a continuous-time spiking neuronal network, we derive the update rule for the synaptic weight distribution and call it the Synaptic Filter. The Synaptic Filter is computationally relevant because it outperforms a gradient rule with optimised learning rate parameter in a dynamic weight estimation task, confirming a previous result obtained in a discrete-time setting and without including weight correlations [48]. Leveraging the continuous-time setting and weight correlation, the Synaptic Filter makes three experimental predictions. First, the mean synaptic weight change depends on the precise timing of pre- and postsynaptic spikes and is therefore reminiscent of spike-timing dependent plasticity (STDP), which yields long term potentiation (LTP) of the synaptic strength if the postsynaptic spike follows the presynaptic spike (within a certain time window) and long term depression (LTD) otherwise [49, 50]. Normative models of STDP have provided a consistent view of the pre-post LTP lobe [4–7, 9, 13]. Pre-before-post pairs induce LTP and thereby reinforce causality. Therefore the time constant of LTP reflects the EPSP time constant. However, normative models do not provide a unifying view on the LTD window [13]. Here, we provide a novel computational rationale for the LTD lobe, namely to compensate for a change in bias. Secondly, based on the hypothesis that EPSP variability encodes synaptic weight uncertainty [48] we formulate the novel prediction that EPSP variability also changes as a function of the precise timing of the spikes. Finally, weight changes induced by joint pre- and postsynaptic activity at one synapse can induce weight changes at synapses that did not receive presynaptic input, reminiscent of the phenomenon of heterosynaptic plasticity. Indeed, our learning rule can explain the negative correlation between homo- and heterosynaptic plasticity observed in experiments [51].

## 2 Results

### 2.1 The Synaptic Filter

The goal of learning is to find predictive functions from training data $\mathcal{D}$ which map inputs $x$ to outputs $y$, typically based on a parametrised generative model. The generative model specifies how the output $y$ of the predictive function is computed from the parameters $w$ and the input $x$. Accounting for a dynamic environment on the level of the generative model makes the data and the parameter $w$ time dependent. Accounting for the fact that many parameter instantiations are compatible with the training data, learning corresponds to computing predictions based on parameter uncertainty. Thus, a given static generative model for learning can be generalised by including the assumption of a dynamic environment or parameter uncertainty.

Including both, time and parameter uncertainty, yields the framework of learning as filtering. Fig 1A illustrates how filtering generalises the static optimisation approach, which is the dominant learning framework. The learning task of static optimisation is to find a point estimate $w^\star$ in parameter space given the generative model and the data $\mathcal{D}$. By including parameter uncertainty, the learning task generalises to inferring the posterior distribution over parameters $p(w|\mathcal{D})$. In the limit of infinite data and for a convex parameter landscape, the parameter distribution collapses around a point, yielding similar results for parameter optimisation and inference, i.e., $p(w|\mathcal{D}) \approx \delta(w - w^\star)$. However, in many problems equivalent minima exist and the amount of data is limited such that the posterior distribution is not well approximated by a point estimate. Another extension of static optimisation considers dynamically changing environmental statistics, i.e., a time-dependent data distribution. In this case, the learning task is to track the optimal parameter as a function of time $w_t^\star$. In a filtering framework, which includes both parameter uncertainty and time, the task is to compute the so-called filtering distribution over the weights as a function of time $p(w_t|\mathcal{D}_t)$ given all previous observations up to time $t$.

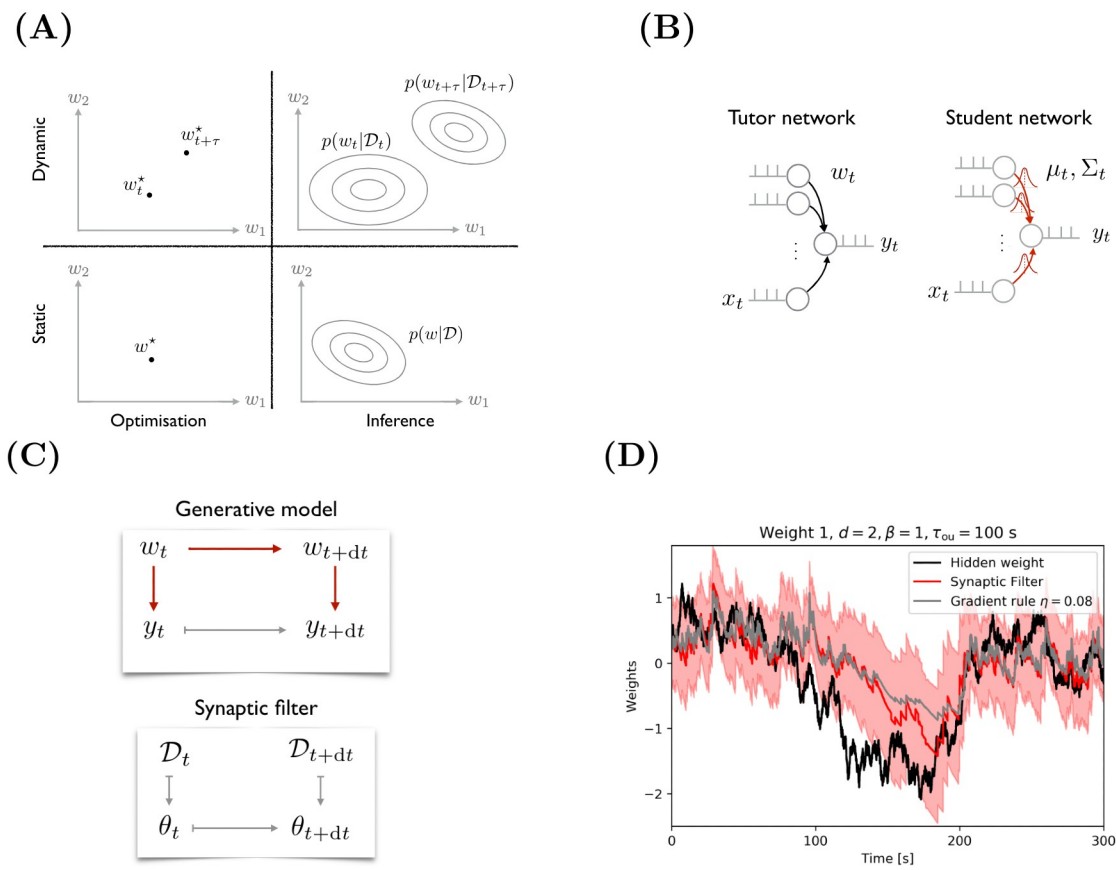

**Fig 1.** **(A)** Learning tasks can be static or dynamic, and deterministic or stochastic. **(B)** The generative model represents the assumption that the observed spike train $y_t$ was generated from a tutor network with the same input $x_t$ and hidden weights $w_t$. **(C)** Graphical model representation of the generative model (top), the Synaptic Filter (bottom) with deterministic dependencies shown in gray and probabilistic ones in red. **(D)** Time series of a ground truth weight (black) in the tutor network and weight distribution (red, shaded area = 2-SD) learned by the student network along with the weight learned by a gradient learning rule (gray) with learning rate $\eta = 0.08$. Between $t = 100$ s and $t = 200$ s, both learning rules fail to closely follow the sharp drop in the hidden weights. In this interval, the estimated uncertainty from the Synaptic Filter is higher which leads to larger update steps and faster learning.

**A generative model to study spike-based plasticity.** To study learning as filtering in the context of spike-based plasticity, we consider a generative model with time-dependent parameters $w_t \in \mathbb{R}^d$, the weights. In contrast to learning as static optimisation, we do not assume that the weights are fixed. Changes in the weights reflect changes in the statistics of the environment. Here, we assume that the weights follow an Ornstein-Uhlenbeck (OU) process with mean $\mu_{\mathrm{ou}} = 0$, diagonal equilibrium covariance matrix $\Sigma_{\mathrm{ou}} = \sigma_{\mathrm{ou}}^2 \mathbb{1}$ with (non-zero elements) $\sigma_{\mathrm{ou}}^2 = 1$ and time scale $\tau_{\mathrm{ou}}$:

$$\mathrm{d}w_t = \tau_{\mathrm{ou}}^{-1}(\mu_{\mathrm{ou}} - w_t)\mathrm{d}t + \sqrt{2\sigma_{\mathrm{ou}}^2\tau_{\mathrm{ou}}^{-1}}\mathbb{1}\mathrm{d}V_t, \tag{1}$$

where $V_t$ is a $d$-dimensional Wiener process. The process in Eq (1) can be represented as Gaussian transition probability: $p(w_t|w_{t-\mathrm{d}t}) = \mathcal{N}(w_{t-\mathrm{d}t} + \tau_{\mathrm{ou}}^{-1}(\mu_{\mathrm{ou}} - w_{t-\mathrm{d}t})\mathrm{d}t, 2\sigma_{\mathrm{ou}}^2\tau_{\mathrm{ou}}^{-1}\mathbb{1}\mathrm{d}t)$. The limit of a large OU time constant represents a static environment while the limit $\tau_{\mathrm{ou}} \to 0$ represents an environment that changes too fast for meaningful learning. The choice of parameters can also be motivated from the perspective of the resulting learning rule. Zero mean yields a weight decay on the time scale $\tau_{\mathrm{ou}}$. Weight decay has been observed experimentally [52]. A diagonal covariance matrix reflects the assumption that weights are not correlated in the

absence of inputs. As we show in section 2.5, one way to study the effect of non-diagonal covariances on learning is by preconditioning with a spiking protocol of highly correlated inputs.

At each moment in time, the weights relate the input spike trains to the output spike of a single neuron via the observation probability $p(y_t|x_{0:t}, w_t)$. For the observations, we assume a Spike-Response Model [53]. The output spikes $y_t$ are generated stochastically from a membrane potential $u_t$ with time constant $\tau_m$ = 25 ms via an inhomogeneous Poisson point process. To connect the membrane potential to the firing rate of the Poisson process, we choose an exponential gain function. Exponential gain functions have been established as a phenomenological model of neocortical pyramidal neurons. They represent a neuron close to the onset of firing but exclude saturation effects of the firing rate at high values of the membrane potential [54]:

$$y_t \sim \text{PoissonProcess}(g(u_t)) \tag{2}$$

$$g(u_t) = g_0 \exp(\beta u_t), \tag{3}$$

where $g_0$ is the baseline firing rate. The determinism parameter $\beta$ controls how strongly changes in the membrane affect the firing rate. For $\beta \to \infty$, the neuron´s firing rate is deterministic. If the membrane potential is below 0, the neuron does not fire; whereas if the membrane potential is above 0, it fires with probability 1. Note that the choice of the exponential gain function is also motivated by the analytical tractability of the filter. For non-exponential gain functions, approximate filters can be obtained by Taylor expansion as discussed in [55].

For the generative model, we assume that the membrane dynamics is much faster than the dynamics of the ground truth weights, i.e., $\tau_m \ll \tau_{ou}$. In this regime, the leaky integration of the weighted sum of input spike trains $x_t$ can be approximated by the weighted sum of the presynaptic activation $x_t^\epsilon := (x * \epsilon)_t$. The presynaptic activation is the convolution of the presynaptic spike train with the exponential kernel $\epsilon_t = e^{-t/\tau_m}\Theta(t)$ and $\Theta(\cdot)$ denotes the Heaviside function. With this we can approximate the membrane potential of the leaky integrator neuron:

$$u_t^{\text{LIF}} = (w^\top x * \epsilon)_t \approx w_t^\top (x * \epsilon)_t = u_t. \tag{4}$$

Optionally, a bias parameter $w_{t,0}$ can be included by setting the first input to unity. The bias controls the excitability of the neuron and will be used in Section 2.3, 2.4 and 2.5. While the leaky integrator $u_t^{\text{LIF}}$ more faithfully represents biological neurons, the approximated membrane potential $u_t$ simplifies the generative model substantially by casting it as a Markov process. Otherwise, the current observation $y_t$ would depend on the entire history of hidden weights through the low-pass filtered membrane potential. The current observation does, however, depend on the history of input spikes via the convolutions $x_t^\epsilon$. This type of history dependency does not complicate the analysis because it can be straightforwardly taken into account in the spiking probability: $p(y_t|x_{0:t}, w_t)$ can be replaced by $p(y_t|x_t^\epsilon, w_t)$. Moreover, it has the biological interpretation of a presynaptic trace. The generative model is represented as graphical model in Fig 1C.

**An Assumed Density Filter solution: The Synaptic Filter.** The generative model can be conceptualised as a tutor network with ground truth weights $w_t$, illustrated in Fig 1B, generating the observed output spikes from given inputs. Learning as filtering corresponds to a student network that continuously computes the distribution over the ground truth weights $p(w_t|\mathcal{D}_t)$ given the history of inputs and outputs $\mathcal{D}_t = \{(x,y)_\tau\}_{\tau=0}^t$.

Generally, the filtering distribution $p(w_t|\mathcal{D}_t)$ is intractable. Here, we obtain an approximated solution with an Assumed Density Filter (see Section C in S1 Text for the derivation), i.e., the exact filtering distribution $p(w_t|\mathcal{D}_t)$ is approximated with the parametric distribution $q_{\theta_t}(w_t) = \mathcal{N}(w_t; \mu_t, \Sigma_t)$ where the distribution parameters $\theta_t \coloneqq (\mu_t, \Sigma_t)$ denote the mean $\mu_t$ and covariance matrix $\Sigma_t$ of the Gaussian. For the proposed generative model, we call the Gaussian Assumed Density Filter the *Synaptic Filter*. An Assumed Density Filter reduces the problem to updating the distribution parameters $\theta_t$ based on observations:

$$\dot{\mu}_t = \beta \Sigma_t x_t^\epsilon (y_t - \gamma_t) + \tau_{\mathrm{ou}}^{-1}(\mu_{\mathrm{ou}} - \mu_t), \tag{5}$$

$$\dot{\Sigma}_t = -\beta^2 \gamma_t (\Sigma_t x_t^\epsilon)(\Sigma_t x_t^\epsilon)^\top + 2\tau_{\mathrm{ou}}^{-1}(\Sigma_{\mathrm{ou}} - \Sigma_t), \tag{6}$$

where $\gamma_t$ is the expected firing rate, i.e., the expected firing rate $g(u_t)$ computed based on the approximated filtering distribution $q_{\theta_t}(w_t)$. It is computed as:

$$\gamma_t \coloneqq \int_{\mathbb{R}^d} q_{\theta_t}(w_t) g(u_t) \mathrm{d}w_t = g_0 \exp\left(\beta \mu_t^\top x_t^\epsilon + \frac{1}{2}\beta^2 (x_t^\epsilon)^\top \Sigma_t x_t^\epsilon\right). \tag{7}$$

The expected firing rate depends not only on the mean of the filtering distribution but also on the covariance. As expected from a convex gain function, the covariance increases the expected firing rate compared to a scenario where only the mean is taken into account.

The first term in Eqs (5) and (6) originates from observations, therefore, it scales with $\beta$. The update of the mean has the structure of the 3-factor learning rule [56] with classical Hebbian factors, i.e., the pre-synaptic activation $x_t^\epsilon$ and the difference between observed and expected output $y_t - \gamma_t$, and the covariance $\Sigma_t$ as a third factor with a modulatory function, which has been linked to the computation of surprise [57]. The second term in Eqs (5) and (6) comes from the hidden dynamics of the weights, which is why it is proportional to the inverse time constant $\tau_{\mathrm{ou}}^{-1}$. For $\beta = 0$ the updates become independent of observations of the environment, i.e., the hidden weights. In general, the covariance update in Assumed Density Filtering depends on the observations $y_t$. However, the combination of a Gaussian filtering distribution and an exponential gain function yields an update (Eq (6)) independent of the observations; an interesting similarity with the Kalman filter.

Fig 1D illustrates that the Synaptic Filter (red) successfully tracks the weights of the tutor network (black). A gradient rule (grey) with a well-chosen but fixed learning rate is shown for comparison. Unlike the Synaptic Filter, it cannot adjust its updates based on the remaining uncertainty around the hidden weight. In Section B1. in S1 Text, we show that the Synaptic Filter is a good approximation of the true filtering distribution.

The derivation of the Synaptic Filter is valid for any Gaussian proposal with a block diagonal covariance matrix (S1 Text in Section C.5). In our analysis, we focus on three variants of the Synaptic Filter. The *Full* Synaptic Filter, as introduced above, follows from a proposal distribution with a single block covariance of size $d \times d$. The other extreme is the *Diagonal* Synaptic Filter, with $d$ blocks of size $1 \times 1$. The *Block* Synaptic Filter lays between these extremes with $m$ blocks of size $b = d/m$. Different block sizes correspond to different trade-offs between model flexibility and computational demands. We took $b = 8$.

## 2.2 MSE performance of the Synaptic Filter

The natural performance metric in filtering is the normalised Mean Square Error (MSE). Other performance metrics, such as the central moments of the filtering distributions are studied in the S1 Text B.1 and displayed in S2 Fig. It quantifies how closely the mean of the filtering

distribution follows the weights of the tutor, compare Fig 1B. The MSE is defined by $\text{MSE} := d^{-1}\langle(w_t - \mu_t)^\top(w_t - \mu_t)\rangle_t$ where $w_t$ denotes the teacher's weight and $\mu_t$ is the best estimate of the Synaptic Filter. In the case of the gradient rule, $\mu_t$ is replaced with $\hat{w}_t$, the best estimate of the weight.

In the following, the MSE performance of the Synaptic Filters is evaluated for a range of values of the determinism parameter $\beta$, the input firing rate $\rho$ and the number of input synapses, i.e. dimension $d$. Additionally, we compare the MSE of a gradient rule with an optimised learning rate against the Synaptic Filters.

As a benchmark for the Synaptic Filter, we use a gradient rule with a scalar learning rate (see Section 4.1.2 in Materials and methods). This implicitly defines a Euclidean metric in weight space. A more general learning choice corresponds to a matrix-valued learning rate. As shown in Fig 2A, the MSE of the gradient rule (grey) is higher than the MSE (red) of the Synaptic Filter for a large range of learning rates. The MSE as a function of the learning rate $\eta$ exhibits a minimum when the combined effect of delayed learning and overshooting is minimal (see S1 Text in Section B.1 as well as S3 Fig for the computation of the optimal learning rate). Delayed learning occurs at low learning rates because the gradient does not converge before the ground truth weights change. At high learning rates, the update steps of the gradient rule are too large which leads to overshooting. In contrast, the Synaptic Filter optimally tunes the learning rate for each weight individually and at each moment based on the amount of information available in the data.

The MSE of all filtering models decreases as a function of $\beta$, as shown in Fig 2B. The Synaptic Filter (red line) performs slightly better than its diagonalised (black line) or blockwise (blue line) counterparts. As $\beta$ increases, the observations convey more information about the ground truth weights $w_t$, hence allowing for a more accurate estimation. At $\beta = 0$, the MSE of all models is equal to 1, a value that corresponds to the MSE of the prior.

When the dimensionality of the input increases, the Diagonal Synaptic Filter, the Block Synaptic Filter and the Full Synaptic Filter have an increasing number of parameters due to the increasing number of non-zero elements in the covariance matrix. Whether or not the additional parametric complexity yields gains in performance depends on the sparsity of the input. In Fig 2C, the MSE is measured as a function of the input firing rate $\rho$, which is one of the model parameters that controls the sparsity. In the sparse regime, only one input is active in the time window $\tau_\text{m}$, i.e. $\rho\tau_\text{m} d \to 0$. Correlations of weights represent the uncertainty introduced by the co-occurrence of inputs. Thus, representing weight correlations does not improve the performance of Block and the Full Synaptic Filters compared to the Diagonal Synaptic Filter, which cannot represent weight correlations. As shown in Fig 2C, all learning rules have the same MSE in the sparse regime, i.e., if $\rho$ is small. However, as $\rho$ increases, the inputs become non-sparse and input correlations become more frequent. Consequently, the capacity to represent weight correlations matters more. The ranking of learning rules according to their performance follows their ability to represent weight correlations. The Full Synaptic Filter performs best, followed by the Block Synaptic Filter, and the Diagonal Synaptic Filter performs worst. Here, the level of sparsity is controlled by $\rho$. Alternatively, the inputs can be less sparse by increasing the time constant $\tau_m$ such that presynaptic kernels overlap more.

Biological neurons receive thousands of inputs. Therefore, it is important to study how the performance of the learning rules scales as this dimension increases. To make performance simulations comparable across dimensions, the expected firing rate of the output neuron and its variance are kept constant. This is achieved by making the determinism parameter inversely proportional to the square root of the dimension: $\beta \propto d^{-1/2}$. As depicted in Fig 2D, the MSE of all four learning rules increases as a function of the dimension until it saturates at the MSE of the prior, i.e. $\sigma_\text{ou}^2 = 1$. Learning becomes harder because the information about an increasing

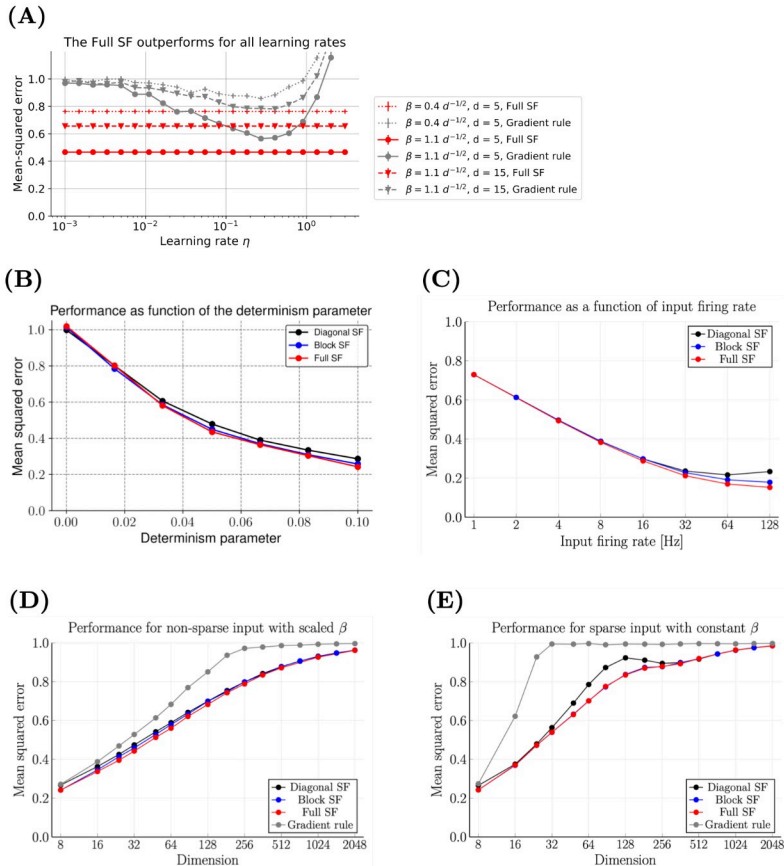

**Fig 2. The Full Synaptic Filter has the best overall MSE across a range of parameters. (A)** The MSE of the Full Synaptic Filter (red) is lower than the MSE of a gradient learning rule (grey) for a range of learning rates $\eta$. The symbols indicate three combinations of determinism $\beta$ and dimension $d$. Consistent with **(B, D)**, the lowest MSE (black) is obtained at the lowest dimension and highest determinism. **(B)** The MSE of Full Synaptic Filter (red line) and the Diagonal Synaptic Filter (black line) decrease as the determinism $\beta$ increases. At $\beta = 0$, the MSE corresponds to the equilibrium variance of the prior, $\sigma_{ou}^2 = 1$. The Diagonal Synaptic Filter performs slightly worse. **(C)** Increasing the input firing rate $\rho$ decreases the MSE and increases the difference between the variants of the SF. The reason for the lower MSE is that the output neuron fires more frequently, thus more information is available for learning at the synapses. The MSE differences between the SF variants are caused by an increase in correlations between the inputs, and hence correlations in the weights. The full SF is best able to captures correlations while the diagonal SF does worst. The block SF falls in between these two. **(D)** The MSE of all learning rules increases monotonically with the number of dimensions until it saturates at $\sigma_{ou}^2 = 1$. The variants of the SF perform better than the optimal gradient rule, particularly in high dimensions. The MSE difference between the SF variants are qualitatively similar to **(C)**, particularly visible in the interval $16 \leq d \leq 128$. For comparability, the expected output firing rate is kept constant by scaling the slope/determinism parameter $\beta \propto d^{-1/2}$. **(E)** With sparse inputs, the performance of all learning rules drops. The optimised gradient rule fails to learn for $d > 32$. The performance of the block and the full SF is identical because the sparsity of the inputs induces correlations only in the blocks of the covariance matrix that both learning rules share. The diagonal SF performs worst because it cannot capture the correlations. For $d > 256$, the performance of SF variants is indistinguishable again.

number of hidden weights is contained in the same number of observed output spikes. The Synaptic Filters outperform the gradient rule by a large margin. The performance differences between the Synaptic Filters are small but consistent for $d < 128$. They resemble the ordering found before: the Full Synaptic Filter outperforms the Block Synaptic Filter and the Diagonal Synaptic Filter is worst. The performance of all learning rules improves when increasing the amount of available information about the hidden weights, e.g., by increasing $g_0$ the firing rate of the output neuron directly or indirectly via a higher input firing rate $\rho$.

To study the performance in high dimensions, we have made two assumptions: first, that the input firing rates are homogeneous and constant, and, secondly, that the average output firing rate of the neuron is kept constant via the scaling of determinism parameter $\beta$. Another way to get a consistent scaling with respect to the input dimensionality is to have a block-sparse structure in the input, referred to as *sparse* for short, i.e., at any time there are only $b$ neurons active. More precisely, the input neurons are divided into blocks of size $b = 8$ and activated one block at a time for the duration of $\tau_{\text{block}} = 1$ s. This block-sparse input structure is a simple implementation of the fact that inputs that target the same dendritic branch are highly correlated [58]. With one block active at any point in time, the expected output firing rate does not depend on the total number of blocks, i.e., the total dimensionality. Thus, in contrast to the previous simulations, there is no need to scale $\beta$ to keep the output neuron's statistics invariant. Fig 2E shows that the performance of all learning rules drops with increasing dimensions. The loss in performance is more pronounced than in the case of non-sparse inputs. The gradient rule with optimal learning rate performs much worse than in the non-sparse case. The Block and the Full Synaptic Filter have the same performance because both capture correlations within blocks equally well. In contrast, the Diagonal Synaptic Filter cannot capture them and performs substantially worse at intermediate dimensions.

In summary, the performance simulations show that the Synaptic Filters substantially outperform a gradient rule with an optimal learning rate. The performance gains are largest for 30 to 1000 dimensions. The variants of the Synaptic Filter perform equally well when input correlations are rare. However, in the presence of input correlations, the Block and the Full Synaptic Filter outperform the Diagonal Synaptic Filter. This provides a computational rationale for including the off-diagonal elements of the covariance matrix. However, the computational benefit comes at the cost of additional parameters. In the case of the Full Synaptic Filter, the number of parameters scales as $\mathcal{O}(d^2)$, which is prohibitive. In addition, it is unclear how information between spatially distant synapses could be exchanged to learn the corresponding elements of the covariance matrix. The Block Synaptic Filter does not suffer from this problem because the number of parameters scales linearly with the dimension and the (constant) block size $b$: $\mathcal{O}(b \times d)$. The biological implementation of the off-diagonal elements is plausible as long as the correlated inputs target synapses that are spatially close, e.g., on the same dendritic branch. Indeed, there is evidence that activity at neighbouring synapses is more likely to be correlated [58].

## 2.3 The Synaptic Filter is consistent with STDP

Spike-time dependent plasticity (STDP) refers to the property of a synapse to exhibit long-term potentiation (LTP) if the presynaptic spike comes before the postsynaptic spike and otherwise to exhibit long-term depression (LTD) [49, 50]. The results are usually depicted as STDP curve, i.e., the normalised change in synaptic weight as a function of the time interval between the pre- and postsynaptic spike.

Normative models of STDP have explained the LTP lobe, i.e., the weight change for pre-before postsynaptic spiking, in terms of causality reinforcement [4–7, 9, 13]. The delay of the postsynaptic relative to the presynaptic spike represents the degree to which the occurrence of the postsynaptic spike can be attributed to the presynaptic activity trace and its decay resembles the shape of the LTP lobe. In contrast, a post-before-pre spike pair has no causal relationship. Indeed, the computational rationale for the LTD lobe has remained a matter of debate with proposals including the regulation of the postsynaptic firing rate and temporal locality [13].

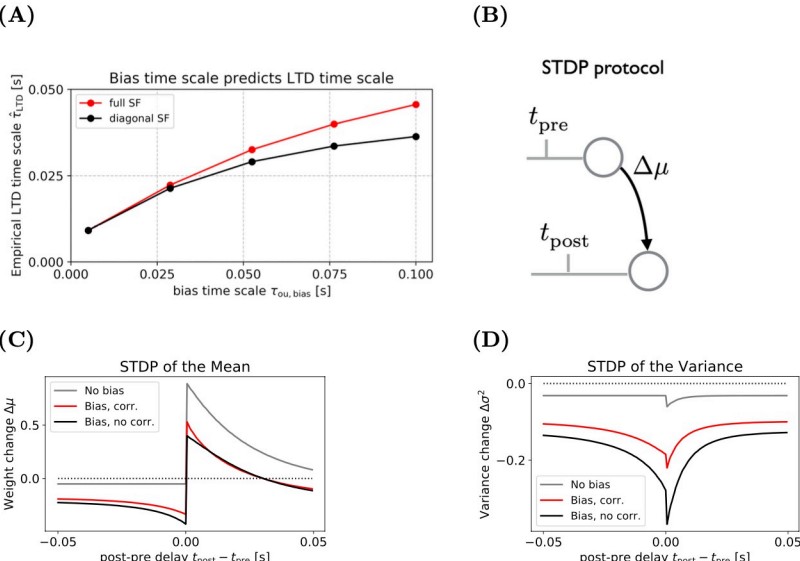

**Fig 3.** **(A)** The empirical time scale of the LTD lobe $\hat{\tau}_{\mathrm{LTD}}$ depends monotonically on the time scale of the bias $\tau_{\mathrm{ou,b}}$. The empirical time scale is defined as post-pre delay that yields 1/e of the amplitude of the negative lobe. With $\Delta\mu(-\infty)$ as zero, it is implicitly defined by the condition $\mathrm{e}\Delta\mu(\hat{\tau}_{\mathrm{LTD}}) = \Delta\mu(0_-)$, where $\Delta\mu(0_-)$ denotes the amplitude of the negative lobe. **(B)** Illustration of the STDP protocol. The weight change $\Delta\mu$ is recorded as a function of the timing difference between the pre- and postsynaptic spike. **(C)** The change of the mean of the filtering distribution as a function of the temporal difference between a pre-post spike pair. For a single weight (excluding the bias, gray line), the Synaptic Filter produces only the LTP lobe ($t_{\mathrm{pre}} < t_{\mathrm{post}}$) while the LTD lobe ($t_{\mathrm{post}} < t_{\mathrm{pre}}$) is independent of spike-timing. Inclusion of the bias, either with off-diagonal covariance elements (red line) or without (black line) yields the LTD lobe; and the magnitude of LTP decreases. **(D)** For the same protocol and learning rules, the variance $\sigma^2$ of the weight exhibits a spike-timing dependent decrease. When the bias (black and red lines) is included, the the change in variance resembles a symmetrised LTD lobe, i.e., it scales as the inverse of $|t_{\mathrm{pre}} - t_{\mathrm{post}}|$. Without the bias, the amplitude of change is reduced and the dependence on spike-timing disappears for $t_{\mathrm{post}} < t_{\mathrm{pre}}$. See Section 4.4 in Materials and methods for simulations details.

We wondered whether the Synaptic Filter could reproduce the STDP curve, in particular the LTD part, without invoking the rare mechanism of self-induced bursting [13]. Specifically, we studied the effect of a time varying bias $w_{t,0}$ to the membrane potential in the generative model. Technically, the bias $w_{t,0}$ is a weight with constant, unit input, i.e., $x_{t,0}^\epsilon \equiv 1$. We use the first index $i = 0$ to represent the bias. From the perspective of a time varying bias, STDP is a secondary (differential) effect, i.e., changes in the synaptic weight account for the prediction error left unexplained by the adjustment of the bias. Immediately after a postsynaptic spike, the expected firing rate is increased, leading to more LTD (Eq (5)). Thus, the time scale of the LTD lobe corresponds to the time scale of the transition probability of the prior $\tau_{\mathrm{ou,bias}}$. Fig 3A shows that the empirical LTD time scale, i.e., the point of 1/e decay of the negative lobe, is the time scale of the bias. In the STDP simulations, we assumed $\tau_{\mathrm{ou,bias}} = \tau_{\mathrm{m}}$ but set all other transition time scales to values much larger than the duration of the experiment (see Section 4.4 Materials and methods). One rationale for assuming $\tau_{\mathrm{ou,bias}} \approx \tau_{\mathrm{m}}$ is that the bias represents the contribution from a set of randomly firing inputs. In the limit of a large number of inputs the bias can be approximated by an Ornstein-Uhlenbeck process. The autocorrelation time of the process is characterised by the presynaptic kernels, i.e., $\tau_{\mathrm{m}}$, and fluctuations of the input rates, which are typically on the order of 100 ms.

To study the effect of the bias on the STDP curve, we applied an STDP protocol with one spike pair to three versions of the Synaptic Filter: a single synapse without bias, the Diagonal Synaptic Filter with bias and Full Synaptic Filter with bias. Assuming that all inputs considered

below are part of the same block, the Full Synaptic Filter and the Block Synaptic Filter are identical, therefore the Block Synaptic Filter is not included explicitly. To avoid effects from transients, the protocol was applied after the bias had reached its steady state. In contrast to biological experiments with up to 60 spike pairs, we simulated a single spike pair to avoid complications from saturation effects and induction times.

In all three experiments, the resulting STDP curve shows an exponentially-shaped LTP lobe but the LTD lobe occurs only when the bias is included, as shown in Fig 3C). The LTP lobe mirrors the exponential shape of the presynaptic activation because when the postsynaptic spike occurs, the dominant part of the weight update is proportional to the EPSP amplitude $\epsilon$ (see the first term on the RHS of Eq (5). The LTD lobe is present when the bias is included (black and red lines). Without the bias (grey line), the LTD part of the STDP curve is independent of the spike timing. Moreover, the bias lowers the amplitude of the LTP. Both observations, the LTD lobe and the lower LTP are caused by the modulation of the expected firing rate $\gamma_t$ due to the bias $w_{t,0}$. The expected bias acts as a low-pass filter of the postsynaptic spike train. Its value is maximal immediately after the occurrence of a postsynaptic spike at $t_{\text{post}}$ and relaxes back to its equilibrium afterwards. The faster the pre follows the postsynaptic spike, the larger the value of the expected firing rate $\gamma_{t_{\text{pre}}}$ when the presynaptic spike occurs. Since LTD is proportional to $\gamma_t$, shorter intervals between pre- and postsynaptic spikes lead to more LTD, in correspondence with biological STDP experiments. The dynamics of the variables of the Synaptic Filter are shown in detail in S4 Fig in Section E of S1 Text.

The Full Synaptic Filter and Diagonal Synaptic Filter are consistent with STDP. The appearance of the LTD lobe is contingent on the inclusion of the bias. Indeed, since the underlying mechanism does not require the inclusion of uncertainty, a similar result could be obtained in a learning as optimisation framework as well, e.g., as the post-only term in the expansion of a generic update function [53]. The fact that only a single synapse with bias was modelled does not impair generality because additional unstimulated synapses do not affect the result.

## 2.4 The Synaptic Filter predicts spike-timing dependent changes of the variance

So far, we have assumed that the posterior mean $\mu_t$ can be measured experimentally as average EPSP amplitude, but we haven't made any assumption on the representation of the posterior variance. The synaptic sampling hypothesis assumes that the posterior variance $\sigma_t^2$ corresponds to the EPSP variance [59]. This hypothesis has two consequences. First, it affects the membrane potential with this additional source of stochasticity (the sampled weight) and therefore impacts the form of the Synaptic Filter. In S1 Text in Section A, we show that the performance of this Sampling Synaptic Filter is similar to the performance of the Synaptic Filter (see S1 Fig). Secondly, the synaptic sampling hypothesis affects the biological predictions. So we wondered whether the Synaptic Filter predicts spike-timing dependent changes in the EPSP variance.

Studying the same three conditions as in the previous section, we found that the variance decreases for all conditions and all pre-post timings, as shown in Fig 3D. In a Bayesian framework, this is expected because input spikes, which represent informative data, decrease uncertainty. Interestingly, the reduction depends on the spike-timing. For a single synapse without bias (gray line), the effect is weak and confined to the causal pairings, i.e., $t_{\text{pre}} < t_{\text{post}}$. Including the bias (black and red lines) increases the amplitude of the variance change. Moreover, it adds a qualitatively new feature: a negative lobe in the regime $t_{\text{pre}} < t_{\text{post}}$. The underlying mechanism is the same as in the case of the LTD lobe of the mean (Fig 3A). Changes in the synaptic weight and the bias modulate the expected firing rate $\gamma_t$. In the models with bias, a postsynaptic spike increases the expected firing rate temporarily and, hence, the potential for variance

reduction when a presynaptic spiking occurs close in time. When both spikes coincide $t_{pre} = t_{post}$ the reduction is maximal because the temporary increased bias and the presynaptic activation increase the expected firing rate superlinearly. Thus, with the sampling hypothesis, the Synaptic Filter makes the novel prediction of spike-timing dependent changes of the EPSP variance.

## 2.5 The Synaptic Filter explains heterosynaptic plasticity

From a Hebbian perspective on plasticity, it is required that the presynaptic neuron's activation takes part in the postsynaptic neuron's activation. Heterosynaptic plasticity contradicts a purely Hebbian view on learning because plasticity occurs without presynaptic activation [51, 60]. For example, LTP induction at hippocampal synapses leads to LTD at synapses that did not receive stimulation [61]. It has been argued that the role of heterosynaptic plasticity is complementary to homosynaptic Hebbian plasticity, which can destabilize neuronal dynamics through run-away weights [62, 63].

We wondered whether the Synaptic Filter is consistent with heterosynaptic plasticity. Our starting point was that heterosynaptic plasticity could be linked to the explaining-away effect in Bayesian reasoning. Explaining-away occurs when one computes the posterior over multiple causes for a single observation. When additional observations provide evidence for only one cause, the competing causes are "explained away". For example, hearing a triggered alarm (an observation) is best explained by a burglar. However, upon learning that an earthquake occurred when the alarm was set off, the posterior probability for the burglar decreases.

In the spirit of explaining-away, we designed a preconditioning protocol to set up two synapses as competing causes for observations, i.e., the postsynaptic activity. The preconditioning protocol consists of synchronous spike trains at both inputs without postsynaptic spiking. In a second step, an STDP protocol was applied to the first synapse but plasticity was reported from both. The weight change at the first and second synapse is our prediction for homo- and heterosynaptic plasticity respectively. Both protocols are shown in Fig 4A. We obtained the homo- and heterosynaptic STDP curves with and without preconditioning. The effect of preconditioning is illustrated in Fig 4B: the equilibrium weight distribution, which is the initial condition of the STDP-step, becomes negatively correlated. We simulate a 3-dimensional Full Synaptic Filter with two synapses and bias. To test whether correlations between weights are important for heterosynaptic plasticity, we also include the Diagonal Synaptic Filter. The same time constants and initial conditions are assumed as in the STDP experiment (Section 4.4 Materials and methods).

The homosynaptic STDP curve (black lines) appear robustly in all experiments, as shown in Fig 4C and 4D, i.e. with and without preconditioning (dashed vs solid lines) and in both models, the Full Synaptic Filter and Diagonal Synaptic Filter. Preconditioning lowers the overall amplitude of the STDP curve, which was to be expected because presynaptic activity reduces the variance which acts as learning rate. Only a single experiment yields the heterosynaptic STDP curve: the Full Synaptic Filter with preconditioning (solid red line), shown in Fig 4D (for an illustration of the dynamic variables involved in this heterosynaptic plasticity experiment, see S5 Fig). The heterosynaptic curve has the same shape as the homosynaptic STDP curve but with an opposite sign. In contrast, the Diagonal Synaptic Filter exhibits no heterosynaptic STDP, i.e., the red curves in Fig 4C are flat; and the Full Synaptic Filter without preconditioning has a flat heterosynaptic STDP curve (dashed red line, Fig 4D) as well. These results demonstrate that the mechanism of heterosynaptic plasticity in the Full Synaptic Filter requires weight correlations. The Diagonal Synaptic Filter cannot represent weight correlations, which is why it never exhibits heterosynaptic plasticity. The Full Synaptic Filter, in

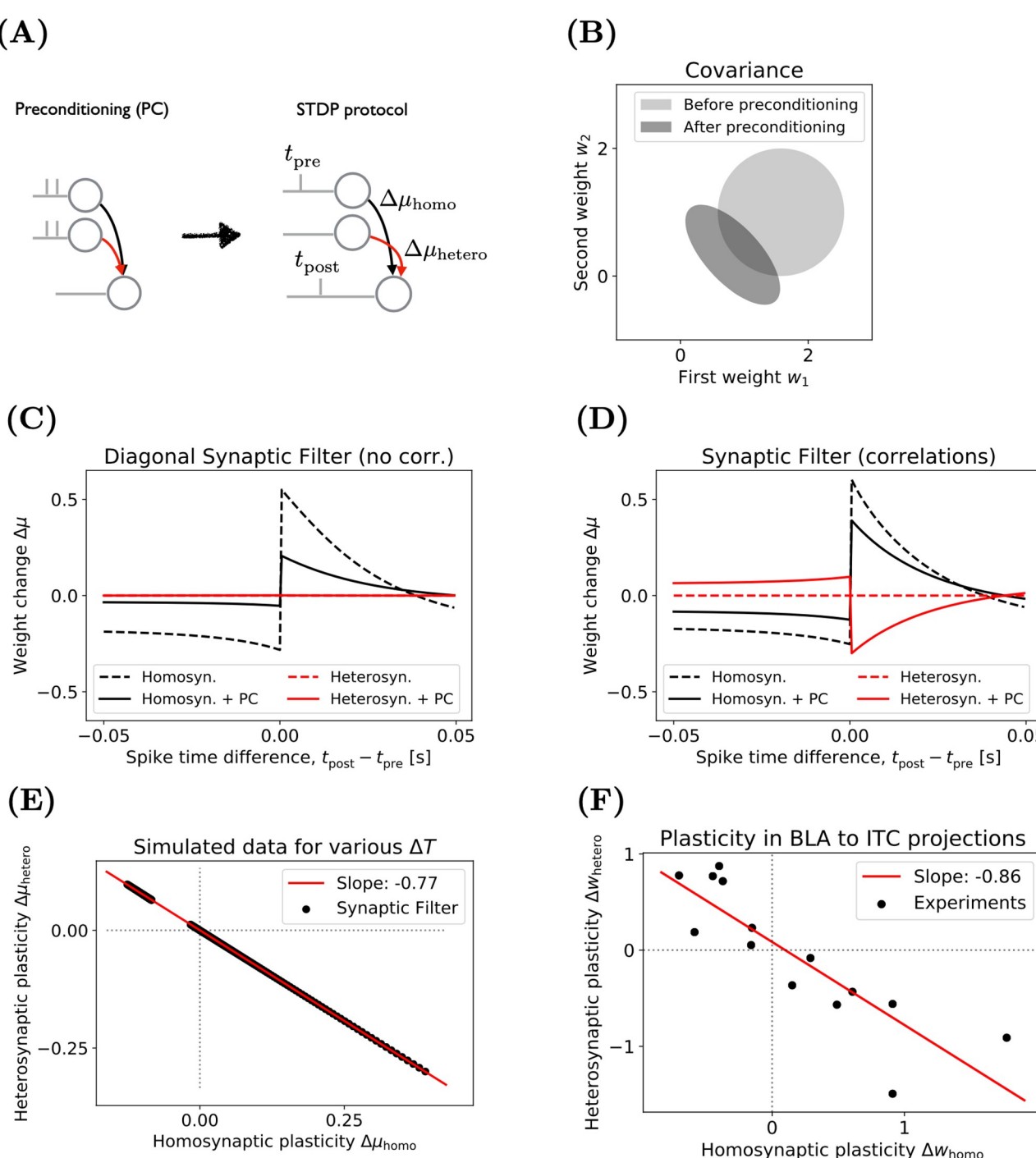

**Fig 4. The Full Synaptic Filter explains experimentally observed anticorrelation of homo- and heterosynaptic plasticity. (A)** Two inputs drive a neuron. During (optional) preconditioning (PC), two synchronous input spikes are delivered. Changes in the weights in the first and second weight in response to an STDP protocol are reported as homosynaptic (black) and heterosynaptic (red) plasticity respectively. See Section 4.4 in Materials and methods for more details. **(B)** The effect of PC on the equilibrium weight distribution, visualised by contours, of the Full Synaptic Filter (light grey). After PC, the weight distribution (dark grey) has lower mean and weights become anticorrelated. **(C)** The Diagonal Full Synaptic Filter exhibits homosynaptic STDP (black lines) but no heterosynaptic plasticity (red lines). PC reduces plasticity (solid black line). **(D)** The Full Synaptic Filter exhibits homo- and heterosynaptic plasticity (solid black and red lines) after PC. Without PC (dashed lines), the Full Synaptic Filter behaves like its diagonal counterpart shown in **C**. **(E)** The Full Synaptic Filter predicts that homo- and heterosynaptic plasticity are anticorrelated. The $x$- and $y$-locations of the black points correspond to the solid black and red lines in **(C)**. **(F)** Anticorrelated homo- and heterosynaptic plasticity was found in synaptic projections from BLA to ITC neurons, Figure redrawn from [51].

contrast, can represent weight correlations but they have to be induced by the preconditioning protocol (because we made the idealised assumption that the equilibrium distribution has strictly uncorrelated weights). The correlations are encoded in the covariance matrix $\Sigma_t$ as off-diagonal elements. The covariance matrix acts as an inverse metric of the parameter space. It scales the update of the mean weight via $\Sigma_t x_t^\epsilon$ [64]. Thus, activity at one of the weights can lead to plasticity at correlated weights. Therefore the Full Synaptic Filter exhibits heterosynaptic plasticity only in combination with the preconditioning protocol.

The observation that homo- and heterosynaptic STDP curves have opposite signs is explained by the negativity of the off-diagonal entries in the covariance matrix. From a mathematical perspective, the sign of the updates of the off-diagonal elements is negative when correlations are present. This follows from using non-negative inputs (see Section 4.2 Materials and methods). Intuitively, the negative correlation between two weights (with same-sign inputs) encodes how much they can explain-away each other.

Next, we wondered whether the negative correlation between homo- and heterosynaptic plasticity was consistent with experimental data. We quantified their relation by first plotting the values of the homo- and heterosynaptic STDP curves against each other and subsequently fitting a line, shown in Fig 4E. The negative slope means that homosynaptic LTP is correlated with heterosynaptic LTD. The amplitude of heterosynaptic plasticity is around three-quarters of the amplitude of homosynaptic plasticity. While the value of the slope depends on the number of spikes in the preconditioning protocol and other model parameters, the negativity of the slope is a robust feature of the Full Synaptic Filter caused by the negativity of the weight correlations. A similar relation between homo- and heterosynaptic plasticity was reported in projections from the basolateral amygdala (BLA) to intercalated cells (ITC) of the amygdala [51]. The authors used extracellular low- and high-frequency stimulation to induce LTD and LTP respectively. They associated the induced weight change in the stimulated connection with homosynaptic plasticity and weight changes at other connections with heterosynaptic plasticity. Their main result aggregates plasticity results from multiple recorded ITC cells in a single figure, replotted in Fig 4F. The data confirms a robust negative correlation between homo- and heterosynaptic plasticity, consistent with the prediction of the Full Synaptic Filter. Fig 4E and 4F are comparable since in both cases the spread of points originate from the variability in the induction mechanism. In the case of the Full Synaptic Filter (Fig 4E, only one parameter, the spike-timing differed between points. In the biological plasticity experiment (Fig 4F), the number of sources of variability is much higher, including somatic properties, initial conditions of synapses, speed of signal transmission and effectiveness of extracellular stimulation. On the premise that the aggregated outcome of this variability modulates the effectiveness of biological plasticity similarly as spike-timing in the simulated experiment, Fig 4E and 4F provides evidence for the Full Synaptic Filter (or the Block Synaptic Filter). Moreover, the Full Synaptic Filter explains the inverse correlation between homo- and heterosynaptic plasticity in terms of the explaining-away effect. The inverse correlation has also been found in the Hippocampus [61] and more generally, see [65] for a review on heterosynaptic plasticity.

## 3 Discussion

In this study, we showcased the framework of learning as filtering through the Synaptic Filter, an Assumed Density Filter for the weights in a spiking network. The main advantage of learning as filtering is that it accounts for the dynamics of the environment and weight uncertainty in a mathematically principled way. In a dynamic learning task, the Synaptic Filter outperforms a gradient rule with an optimised learning rate in weight space. The relevance of the Synaptic Filter to biological plasticity is threefold. It exhibits the STDP of the mean weight,

including the negative lobe; it predicts the STDP of the weight variance; and based on weight correlations, it predicts heterosynaptic LTD and homosynaptic LTP consistently with experimental evidence. Thus, the Synaptic Filter combines computational benefits with biological insights into plasticity.

The framework "learning as filtering" can be used to derive additional learning rules. Here, we considered the simple case of a Gaussian weight distribution, an OU-process as prior and an exponential gain function in combination with spike-based observations. Alternatives yield new learning rules. Parameterising the weight distribution through the binomial model of stochastic release represents an exciting possibility to study dynamics of EPSP variability and quantal parameter plasticity. The case of log-normally distributed weights has been studied in discrete time [59]. From a biological perspective, log-normal or binomial models have the advantage of obeying Dale's law. However, this advantage comes at the cost of intractability, the requirement of additional approximation or more complicated update rules. To avoid these complications, we chose Gaussian synapses in this work. Another option is to study gated plasticity through a hierarchical weight distribution in which an additional hidden variable infers whether a synapse should be plastic or not. Moreover, the framework can encompass different types of observations, for example, continuous rates instead of spikes. Closely related to the observed variable is the choice of the gain function. While an exponential function offers simplicity, the sigmoidal and soft-max functions yield analytically tractable learning rules under additional approximations. Thus, learning as filtering offers a rich set of options to study learning and synaptic learning in particular.

The generalisation of the single neuron analysis to the case of a recurrent neuronal network is straightforward as long as all neurons in the recurrent population are visible (i.e. their activity is prescribed). Indeed, when all neurons are visible, then the likelihood nicely factorises along the temporal dimension because of the chain rule and along the spatial dimension because of the conditional independence between the neurons (i.e the current spiking of neuron $i$ at time $t$ is only affected by the spiking history of all neurons up—but not including—time $t$). In the presence of hidden neurons, this factorisation is not possible anymore for the marginal likelihood. Thus, the Synaptic Filter can be used to gain insights into learning dynamic weight distributions not only in a single neuron but also in the more complex setting of a recurrent neuronal network model, given that all neurons are visible.

For the derivation of the learning rules, we assume that the output neuron receives an external and neuron-specific supervision signal. Recent studies have addressed the question of how such a signal could be computed in biological networks [66] and in continuous time [67]. In our model, the supervision signal takes the form of spikes of the output neuron. This assumption does not exclude the possibility that biological neurons receive one type of spike as a supervision signal to guide plasticity, and generate another type of spike to make predictions [7, 68]. Experimental findings in the cerebellum and the cortex are compatible with this idea. Indeed, complex and simple spikes in Purkinjee cells, and bursts and normal firing in cortical pyramidal neurons play distinct roles for plasticity [69, 70]. Therefore, the assumption of a continuously provided, event-based supervision signal does not impair the biological relevance of the Synaptic Filter.

The Synaptic Filter (except for the diagonal version) represents correlations between weights. From a biological perspective, this suggests two directions for future research. First, is the SF biological plausible? It should be recalled that the Synaptic Filter (and all its variations) is a normative learning rule derived from some computational principle and does not predict in itself how the implementation should be. Indeed, in Marr's 3 level perspective, a computational principle can be achieved by many algorithm and every algorithm can be implemented in different ways, so the bottom-line is that the implementation is not unique. If the

implementation of the learning rule follows exactly the Eqs (5) and (6) it is hard to see how this is biologically plausible. Indeed, taken as such every synapse $i$ depends on the activities of all other inputs in a non-trivial way which seems to violate any sense of locality—which would be desirable for a biologically plausible learning rule. It is however possible to have other implementations that are consistent with Eqs (5) and (6) (or at least closely approximate those equations) and yet have a higher degree of biological plausibility. This would be for example the case for the constant Synaptic Filter which assumes that the inputs are roughly constant, i.e. $x^\epsilon \simeq \bar{x}$ (see S1 Text, Section F). In this case, we can define a surrogate variable $z = \Sigma\bar{x}$ such that synapse $i$ depends only on $z_i$ whose dynamics depends only on itself as well as a global factor $z^T\bar{x}$ which is available at all synapses. Further research is therefore required to determine which implementation best matches biophysical constraints while keeping the end-effects as predicted by Eqs (5) and (6). Secondly, the Synaptic Filter was derived under the assumption of positive presynaptic inputs while the sign of the weights could either be positive or negative. As a consequence, weight correlations are never positive. Based on the negative weight correlations the Synaptic Filter could explain the negative correlations between homo- and heterosynaptic plasticity (shown in Fig 4). As an extension of this result, it would be interesting to generalise the Synaptic Filter to the case of positive and negative inputs, representing excitatory or inhibitory neurons. We hypothesise that a generalised Synaptic Filter would predict positively correlated homo- and heterosynaptic plasticity between synapses from inhibitory and excitatory neurons.

Because uncertainty controls the speed of learning, the Synaptic Filter in combination with the sampling-hypothesis can link synaptic variability to synapse-specific metaplasticity, which has been observed experimentally [65]. The Synaptic Filter predicts that synaptic variability and learning speed are reduced upon presynaptic stimulation but relax back to maximal value on the time scale of the OU-prior. Indeed, consistent with an OU time scale of hours, plasticity experiments have shown that LTP saturates temporally but recovers within hours [71].

The presented work is closely related to the Know-Thy-Neighbour (KTN) theory [72]. It assumes that synapses estimate the presynaptic membrane potential from the arriving spike train within the filtering framework. Consequently, it links the dynamics of short-term synaptic depression to the mean and variance updates of the filtering distribution. The Synaptic Filter and the KTN theory formalise plasticity via Assumed Density Filtering with an Ornstein-Uhlenbeck prior. At any point in time, the synapses encode a posterior distribution over the hidden variable given past observations, i.e., membrane potential and presynaptic spikes in the KTN model and ground truth weight with pre- and postsynaptic spikes in our model. In both models the classically defined synaptic efficacies, the averaged postsynaptic responses, correspond to the mean of the posterior; and the variance plays the role of a learning rate in the update equations. Our novel contributions are the extension of the KTN framework to multiple and potentially correlated hidden variables and a more complex emission process, i.e., the emissions are generated by the sum of the hidden variables weighted by the presynaptic trace. The KTN model is formally equivalent to the one-dimensional Synaptic Filter when the bias is the only hidden variable. On the level of the biological interpretation, KTN focuses on short-term plasticity while our work makes a connection to experiments concerning long-term plasticity.

A previous study has addressed the computational role of synaptic uncertainty [38]. The authors propose that spine turnover implements samples from the posterior distribution over synaptic weights via Langevin sampling. Their work differs from ours because they use a static inference task (bottom right in Fig 1A), not filtering. One consequence of the static nature of their task is that the online version of their learning rule only allows for a fixed data set size as external parameter. Compared to previous learning rules in the context of filtering [59], we

make four contributions. First, we connect the learning task to the rich literature of filtering. In particular, this facilitates a simple, rigorous and continuous-time treatment. Secondly, we go beyond the assumption of a diagonal Gaussian Assumed Density Filtering by including weight correlations and show that their importance for the filtering performance. Finally, based on the spiking, continuous-time analysis, the Synaptic Filter recovers the phenomenon of spike-timing dependent plasticity, i.e., the mean synaptic weight increases if the postsynaptic spike follows the presynaptic spikes closely, and decreases if the spike order is reversed. Moreover, the Synaptic Filter predicts spike-timing dependent changes of the EPSP variability. Finally, it explains the negative correlation between homo- and heterosynaptic plasticity in terms of the Bayesian explaining-away effect.

Overall this article provides evidence that learning as filtering is a promising candidate for a computational principle underlying plasticity and provides testable predictions.

## 4 Materials and methods

### 4.1 The generative model and learning rules

In this section, we define the filtering problem in terms of a generative model for plasticity. In addition, the update equations of the learning models are introduced, i.e., the Synaptic Filter and the gradient learning rule.

**4.1.1 The Synaptic Filter: Update equations and prediction.** The goal of learning as filtering is to compute the distribution over the hidden weights $p(w_t|\mathcal{D}_t)$ given all previously observed input and output spikes $\mathcal{D}_t \coloneqq \{x_\tau, y_\tau\}_{\tau=0}^t$. On a formal level, the Markovian structure of the generative model (under the assumption of Eq (4)), enables a recursive solution of the filtering problem:

$$p(w_t|\mathcal{D}_t) \propto p(y_t|x_t^\epsilon, w_t) \int_{\mathbb{R}^d} p(w_t|w_{t-\mathrm{d}t}) p(w_{t-\mathrm{d}t}|\mathcal{D}_{0:t-\mathrm{d}t}) \mathrm{d}w_{t-\mathrm{d}t}. \tag{8}$$

The Kushner-Stratonovich Equation [44] gives a formal solution for all moments of the filtering distribution. However, for most generative models the solution is intractable because of the closure problem, i.e., the evolution of lower moments depends on higher moments of the filtering distribution. One way to address the closure problem is Assumed Density Filtering. The central idea is to replace the exact filtering distribution with a proposal distribution $q$ parameterised by $\theta_t$:

$$p(w_t|\mathcal{D}_t) \approx q_{\theta_t}(w_t). \tag{9}$$

When substituting the approximation Eq (9) into the right-hand side of Eq (8), the resulting posterior will generally not lay in the family of the proposal distribution $q_\theta$. Therefore, one has to decide how to best approximate the result with a member of the proposal family [73].

Here, we derive the *Synaptic Filter*, an approximate solution based on a Gaussian proposal density with mean $\mu_t \in \mathbb{R}^d$ and covariance matrix $\Sigma_t \in \mathbb{R}^{d \times d}$. The evolution of the distribution parameters $\theta_t = (\mu_t, \Sigma_t)$ can be computed from the Kushner-Stratonovich Equation. To remain in the Gaussian family, higher moments are omitted (see S1 Text, Section C.2). For the generative model specified above, the resulting update equations for $\mu_t$ and $\Sigma_t$ are Eqs (5) and (6).

**4.1.2 Gradient learning rule.** As a performance benchmark, we use the gradient learning rule. Assuming updates are proportional to the gradient of the log output probability yields:

$$\dot{w}_t^{\mathrm{ML}} = \eta \beta x_t^\epsilon \beta (y_t - g_t^{\mathrm{ML}}), \tag{10}$$

where $\eta$ is the learning rate parameter and $g_t^{\mathrm{ML}} = g_0 \exp(\beta (w_t^{\mathrm{ML}})^\top x_t^\epsilon)$. We did not absorb $\beta$ in

the learning rate $\eta$ to make the values of $\eta$ more comparable to values of the variance in the Bayesian learning rule and to use the same scaling of $\beta$ with the dimension as in the Synaptic Filters (see Section 4.3 in Materials and methods).

### 4.2 The weight correlations of the Synaptic Filter are mostly negative

The weight correlations in the Synaptic Filter are represented by the off-diagonal elements of the covariance matrix $\Sigma_t$. In the 2-dimensional case and for positive inputs ($x_t^\epsilon \geq 0$) these elements $\Sigma_{t,i\neq j}$ are always negative. This follows from the following two observations. First, the change of weight correlations is negative when the initial weight distribution is diagonal, i.e., $\Sigma_0 = \sigma_0^2 \mathbb{1}$. Secondly, a negatively correlated weight distribution cannot evolve into a positively correlated weight distribution without assuming a diagonal form in between.

The change of the covariance is given by Eq (6). Omitting the temporal index for clarity and assuming $i \neq j$, the update of an off-diagonal element is:

$$\dot{\Sigma}_{ij} = -\gamma(\Sigma x^\epsilon)_i(\Sigma x^\epsilon)_j - 2\tau_{\mathrm{ou}}^{-1}\Sigma_{ij}, \tag{11}$$

where we used that $(\Sigma_{\mathrm{ou}})_{ij} = 0$. For the initial condition given by a diagonal covariance matrix, this expression simplifies to:

$$\dot{\Sigma}_{ij} = -\gamma\Sigma_{ii}x_i^\epsilon\Sigma_{jj}x_j^\epsilon. \tag{12}$$

Since all factors in this expression are positive but the overall sign is negative, an initially diagonal weight distribution can only evolve towards a negatively correlated distribution. Because in 2 dimensions a transition from a negatively correlated to a positively correlated weight distribution is not possible without a diagonal state in between, positive correlations cannot occur. Conditions under which this result holds for $d > 2$ are discussed in the Section F in S1 Text.

### 4.3 Computational performance: Hyperparameters and simulation details

Here, we describe in detail the hyperparameters and simulations configuration for the results in Section 2.2 except for the details of the results shown in Fig 2A, which are discussed in Section B.3 in S1 Text because they were obtained from a different batch of simulations. The total simulation time is reported in terms of multiples of $\tau_{\mathrm{ou}}$, which we call *epochs*. The initial conditions for all performance simulations were $\mu_0 = \mu_{\mathrm{ou}}$ and $\Sigma_0 = \Sigma_{\mathrm{ou}}$. To remove any dependency on the initial condition, 8 epochs were used as burn-in time. These epochs were not used to compute the mean squared error (MSE).

To compute MSE as a function of the determinism parameter, we used $\tau_{\mathrm{ou}} = 200$ s, $d = 16$ and the input firing rate was $\rho = 40$ Hz. The baseline firing rate for this and the subsequent simulations was $g_0 = 20$ Hz. For the Block Synaptic Filter, two blocks of size 8 were chosen. The time step was dt $= 10^{-3}$ s. After burn-in, the error was averaged over 256 epochs.

The MSE as a function of input firing rate was computed with $\tau_{\mathrm{ou}} = 400$ s and $d = 100$. For the Block Synaptic Filter, 10 blocks of size 10 were chosen. The determinism parameter was set to $\beta = 0.1$, which required a time step of dt $= 10^{-5}$ s. After burn-in, 32 epochs were simulated.

The MSE as a function dimension was computed with $\tau_{\mathrm{ou}} = 200$ s, dt $= 10^{-3}$ s and $\rho = 40$ Hz. The block size for the sparse input and for the Block Synaptic Filter was 8. The determinism parameter was $\beta = 0.01 n_b^{-1/2}$, with $n_b$ the number of blocks. After burn-in, 32 epochs were simulated.

### 4.4 Biological predictions: Parameters and technical details

In this section, we specify the values of the hyperparameters, protocols, initial conditions and simulation parameters used in the simulated STDP experiments in Sections 2.3 to 2.5.

**4.4.1 Hyperparameters.** For the simulated experiments, the membrane time constant and baseline firing rate was set to $g_0 = 1$ Hz and the determinism parameter to $\beta = 1$.

The time scale of the weights and weight correlations was $\tau_{\mathrm{ou}} = 10^4$ s. Because the bias time scale is much shorter, correlations between bias and other weights decay on the order of $\tau_{\mathrm{m}}$ (see S1 Text, Section C.4, Equation S29). To prevent run-away dynamics of the bias in the absence of spikes, we chose $\mu_{\mathrm{ou},0} = 1$. The prior variance $\sigma^2_{\mathrm{ou},0}$ determines how strongly the bias changes in response to input and output spikes. For the STDP experiments in Fig 3, we chose $\sigma^2_{\mathrm{ou}} = 2$ and for the heterosynaptic experiments in Fig 4, we chose $\sigma^2_{\mathrm{ou}} = 1$.

**4.4.2 Protocols and initial conditions.** The STDP protocol consisted of pre- and postsynaptic spikes with 200 different values for the delay (shown on the $x$-axis of the STDP curve). The preconditioning protocol consists of a presynaptic spike pair with 5 ms delay simultaneously applied to both presynaptic inputs. Prior to applying any protocol, the Synaptic Filter was simulated without any input or output spikes for $T_{\mathrm{wait}} = 6\tau_{\mathrm{m}}$ such that the mean and variance values of the bias converge to their equilibria. The same waiting time was simulated after the preconditioning protocol. The simulated STDP curve was computed based on the value of the weight directly before the application of the STDP protocol and the value of the weight after $2T_{\mathrm{wait}}$. The initial conditions for the distribution parameters of the Synaptic Filter were chosen $\mu_{0,i} = 1$ with $i \in (0, \ldots, d-1)$ for the weights and $\Sigma_0 = \sigma^2_0 \mathbb{1}$ with $\sigma^2_0 = 1$ for the covariance.

**4.4.3 Technical details: Simulations and fits.** We solved the ODEs of the Synaptic Filters with the Euler method. The time step was dt $= 10^{-4}$ s for the STDP experiments presented in Sections 2.3 and 2.4. Since the preconditioning protocol induces sharp decreases of the variance, a time step of dt $= 10^{-5}$ s was used for the simulations in Section 2.5 to ensure that the variance remained positive.

The slopes reported in Fig 4E and 4F correspond to a linear fit with a least-squares objective and two free parameters, slope and offset. The data shown in Fig 4F were extracted manually.

## Supporting information

**S1 Text. Additional simulations and derivations.** S1 Fig. The Sampling Synaptic Filters have similar MSEs to their deterministic counterparts. S2 Fig. The first and second moments of the Synaptic Filter match the corresponding moment of the exact filtering distribution. S3 Fig. Optimisation of the learning rate for the gradient rule. S4 Fig. The dynamics of the variables of the Synaptic Filter during the STDP protocol. S5 Fig. The dynamics of the variables of the Synaptic Filter during the heterosynaptic protocol.
(ZIP)

## Acknowledgments

We thank Peter Latham and Máté Lengyel for valuable discussions.

## Author Contributions

**Conceptualization:** Jannes Jegminat, Jean-Pascal Pfister.

**Data curation:** Jannes Jegminat.

**Formal analysis:** Jannes Jegminat, Simone Carlo Surace, Jean-Pascal Pfister.

**Funding acquisition:** Jean-Pascal Pfister.

**Investigation:** Jannes Jegminat, Jean-Pascal Pfister.

**Methodology:** Jannes Jegminat, Jean-Pascal Pfister.

**Project administration:** Jean-Pascal Pfister.

**Resources:** Jean-Pascal Pfister.

**Software:** Jannes Jegminat, Simone Carlo Surace.

**Supervision:** Jean-Pascal Pfister.

**Validation:** Jannes Jegminat.

**Visualization:** Jannes Jegminat, Simone Carlo Surace.

**Writing – original draft:** Jannes Jegminat.

**Writing – review & editing:** Jannes Jegminat, Jean-Pascal Pfister.

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
