## [Decision Letter · Decision Letter 0]

16 Nov 2020

Dear Dr Jegminat,

Thank you very much for submitting your manuscript "Learning as filtering: implications for spike-based plasticity" for consideration at PLOS Computational Biology.

As with all papers reviewed by the journal, your manuscript was reviewed by members of the editorial board and by several independent reviewers. In light of the reviews (below this email), we would like to invite the resubmission of a significantly-revised version that takes into account the reviewers' comments.

In particular, please be sure to address Reviewer 2's comments about valid comparisons to gradient descent, and Reviewer 3's comments on biological plausibility.

We cannot make any decision about publication until we have seen the revised manuscript and your response to the reviewers' comments. Your revised manuscript is also likely to be sent to reviewers for further evaluation.

Sincerely,

Blake A. Richards

Associate Editor

PLOS Computational Biology

Samuel Gershman

Deputy Editor

PLOS Computational Biology

Reviewer's Responses to Questions

**Comments to the Authors:**

Reviewer #1: Jegminat & Pfister

[Note I used the arxiv version for review as I find the Plos format with separate figures and captions annoying.]

This papers interprets synaptic plasticity as a filtering process, tracking a time-varying generator.

The system tries to infere the synaptic weights of the generator and find that under some condition the learning rule resembles STDP. This is a though-provoking study that would suit well in PlosCB.

It is also a technically compentently carried out study. That said, I have my doubts if the results are really offering new insights to STDP.

The authors correctly state that the depression lobe of STDP has not been well explained.

But a simple explanation could be that biophysically it is an efficient way.

Similarly, heterosynaptic competition is present in many models.

* The paper is reasonably clear, but the split of information into Results, Materials and Methods, and Supplementary Information needs attention. As an indication of the poor organization is the fact the \\tau_m was set to 25 ms was mentioned three times.

* In the setup the generative model is another single neuron with dynamic synaptic weights.

In this case it is possible to derive the optimal update rule.

(As an aside, the role of the zero mean and diagional covariance in the generator should be discussed/explored.)

It was not clear to me how this generalizes where the neuron tries to fit a non-linear model with dynamic parameters, where is perhaps not possible to derive the optimal learning rules.

There must be benchmarks for such systems, I would imagine.

* It also raises the question how the parameters of the learning rule (1) and (2) should be learned when they are not known.

* The time-scale of the STDP depression lobe seems to be related to the time-scale of the bias (t_ou,bias) (which would have been nice to be shown in a plot).

The whole argument mathching STDP to filtering hangs therefor on the rather adhoc assumptions that this is somewhere between 10 and 100ms. This needs more motivation.

* Does the STDP curve depend on firing rate (as has been observed)?

Minor:

======

Particle filtering in mentioned without introduction.

MAP is undefined.

In 2.2 it says that beta=beta_0/d , but in 2.1 beta=beta_0/sqrt(d)

There is an edited volume of proceedings by Haykin from the late 90s(?) connecting filtering to neural networks.

Reviewer #2: Summary and appreciation

Many normative models of learning use optimization of a cost function as the core principle. This framework typically ignores parameter uncertainty and the varying nature of the statistical environment. In this manuscript, the authors propose to employ the framework of nonlinear Bayesian filtering as an alternative. They call their approach the Synaptic Filter. In a nutshell, this approach infers the parameters of a Gaussian Assumed Density Filter (= Synaptic Filter), which is an approximation to the exact filtering distribution that gives the probability of the weights given the data. In so doing, they get updating equations for the mean and the covariance of the Gaussian filter; updates of the mean play the role of a standard learning rule, whereas the covariance updates do not typically appear in models.

Using this framework with a Poisson observation model and a Ornstein-Uhlenbeck state model, the authors reproduce well-known properties of synaptic plasticity (STDP rule and the heterosynaptic plasticity with preconditioning) as well as predicting a novel spike-timing dependent effect, this time on the variance of the synaptic weight. Interestingly, their model also suggests an origin for the LTD lobe in STDP. Overall, the authors claim that the Synaptic Filter both benefits predictive performance and brings out interesting conceptual points.

Assumed Density Filters have been studied in the context of Bayesian neural networks for many years as powerful alternatives to standard gradient descent (e.g., Ghosh…Yedidia, AAAI, 2016 for a recent example). In (computational) neuroscience, similar approaches have been used, for instance in the context of the synaptic sampling hypothesis (Aitchison & Latham, arXiv, 2015). The main contributions of the present manuscript are (1) to study synaptic plasticity in the general context of nonlinear Bayesian filtering and, mostly, (2) to apply this framework to Poisson neurons in order to unveil interesting aspects of synaptic plasticity that have been overlooked when studied in other contexts (namely, from an optimization point of view).

My general appreciation of this manuscript is positive, although I do have some points I would like the authors to address and clarify. The manuscript does belong to Plos Comp Biol for its novel interpretation of the LTD lobe, its description of heterosynaptic plasticity and its discussion of the STDP of variance, as well as for (implicitly) advocating for a more general use of nonlinear filtering when studying synaptic models.

Note that I’m not an expert of nonlinear filtering models and that I’m more used to dynamical and optimization-based models of plasticity.

Major points

(1) Variability of the environment

The authors suggest that one of the drawbacks of (pointwise) optimization-based learning is that the variability of the environment is poorly accounted for. However, I wonder how actually variable is the environment represented by the ground truth weight model (the OU process). Indeed, the ground truth model is a OU process with constant mean and variance, hence stationary. Wouldn’t a time-dependent mean and variance be more appropriate to instantiate a changing environment? Along that same vein, perhaps plotting the weight dynamics of the gradient-based model in Fig. 1D would help clarify the expected dynamical differences between Bayesian inference and “pointwise” optimization.

(2) Unfair comparison with gradient descent

The authors argue that the Synaptic Filter might have a better predictive power than gradient descent (GD). To support this, they studied the predictions and performance under different models. My understand is that SF is more efficient than GD in most cases. However, SF seemingly has indirect “insider information” about the ground truth model, given the selected OU process and the Gaussian Assumed density filter. One would expect that a larger statistical mismatch (a larger KL divergence between the exact filtering distribution and the approximation) would be more problematic for SF, even with matching dimensions of the inputs. Also, when there is a model mismatch in term of the input dimensionality, Fig. 2E suggests that GD will eventually outperform SF (this is not shown in Fig. 2E and thus the comment at line 294 is unsubstantiated). The authors do argue (around line 224) that with sparse inputs the effective input dimension might be smaller than the expected 10000 synapses per neurons in the brain, but their operating of 1 spike/membrane time constant does not account for that. I would suggest to i) explore a higher dimension in Fig. 2E to show the point where GD outperform SF and ii) perhaps testing one sparse-input (a few Hz) model at high dimension (optional, but could be interesting). Showing the prediction for GD in Fig. 1D (as in the above point) could also help clarify the differences between the algorithms.

(3) More equations in main text

I do understand the desire to remove “clutter” from the main text by putting equations in the methods. However, my understanding of the paper did suffer from the lack of precise definitions of some quantities. First and foremost, the bias. Indeed, \\omega_{t,0} appears for the first time at line 203 but it was not defined before. Secondly, the expected firing rate \\gamma_t should also appear somewhere after Eq. 2. So, I would suggest to put in the main text the equation for u (emphasizing the bias) and the expression for \\gamma_t.

Minor points

- The authors should refer to the supplementary figures in the main text where appropriate (e.g., refer to Fig. S4 when discussing STDP). Also, in the materials and methods section, they talk about particle filtering without properly referring to the supplements.

- The authors use a comparatively fast time scale for the bias in the paired plasticity protocol. Since they associate this bias to excitability, which encompasses several aspects of neuron spiking, could they discuss what biophysical mechanisms they have in mind (afterhyperpolarization?) for it?

- Do the authors think this framework could reproduce other known aspects of synaptic plasticity, like the frequency dependence?

- Beta in the exponential gain function is constantly referred to as “determinism” parameter. Since the model is purely stochastic, this name is quite odd. Please consider calling it “slope” parameter (or something of sort) instead.

- Lines 236-237: “a low MSE does not necessitate poor predictions.” What did you mean here (esp. the word necessitate sounds out of place)? Moreover, a high MSE performance implies a low MSE. Please consider clarifying the many instances of mentioning MSE to make sure that no confusion remains.

- Lines 240-242: “In case of […] in detail by the brain.” I don’t think this is the issue here. I would say it is rather that no ground truth for the weights is available, only spiking activity supposed to come from that putative ground truth, whose dimensionality could eventually be known (connectomics) but not in every single neurons.

- I think the authors should “shamelessly” push their own tutorial on nonlinear filtering, especially in the introduction.

- I did not follow the discussion of section 5.6 on the negativity of the weight correlation. Please make sure the equations are correct; it felt like you were considering variances and not the off-diagonal terms of the matrix.

- On line 178, typo in word “Diagnalised”.

- Line 228: “Thus it is possible that learning in many brain systems takes place inside the learning regime.” Sounds like a tautology…

- Line 323: “Our rationale was that postsynaptic spiking predominately affects the bias wt,0, which represents the neuronal excitability. “ Difficult to comprehend on first reading. More details would be welcome.

- I didn’t quite follow the discussion in the paragraph starting at line 511. How is the fact that all recurrent neurons be visible change anything to the conditional independence of their spiking?

Reviewer #3: I thank the authors for their submission. As my knowledge of the synaptic plasticity literature is rudimentary, my contribution as a reviewer will be to assess the technical accuracy of the manuscript and to evaluate it on its own terms rather than in reference to the large body of existing literature on synaptic plasticity.

The authors present the Synaptic Filter as an alternative to the optimization-based approaches for setting synaptic weights in learning tasks. The advantage of the Synaptic Filter is that it provides a principled way to track not just the mean value of the weights, but, unlike typical optimization based approaches, also the uncertainty about those weights. This not only improves performance in settings like regression by allowing averaging over weight uncertainty, it also easily accommodates settings in which environmental parameters change over time, which is of obvious relevance to real neural systems. Despite these clear benefits, the SF is only neuroscientifically relevant if the authors can make a convincing case that it is (a) neurally plausible and could indeed, at least in some reduced form, be implemented in the brain; and that (b) it provides significant advantages over other, perhaps less principled, approaches. Thus far I do not believe the authors have done so.

The authors make three specific claims about the advantage of the SF: That it provides a principled explanation of the LTD lobe of STDP; That it provides a principled explanation of heterosynaptic plasticity; and that it provides a testable prediction for the variance in weight changes in STDP experiments. I will address each of these points in turn, before discussing neural plausibility.

First, the positive: The SF predicts that in STDP settings, the variance of synaptic weight changes should drop as tpost approaches tpre from either direction, along with a principled explanation of why this should be so. I found this interesting, testable prediction to be a highlight of the paper. I merely wondered why, given that the authors had compared their predictions for heterosynaptic plasticity to experimental data, why they had not also done so here to test their prediction? This, while not absolutely necessary in my opinion, would significantly strengthen their paper.

The authors also show that in a simple STDP protocol, the SF can reproduce heterosynaptic plasticity similar to that observed in experiment. The principled nature of the SF really shines here in that it not only reproduces the effect, but also provides an explanation for heterosynaptic plasticity through the negative covariance induced between different synapses by the preconditioning protocol.

The authors also claim that the Synaptic Filter reproduces the LTD lobe of the STDP curve, as show in Figure 3A, and provides a principled explanation for it. While the SF does indeed reproduce the general _shape_ of the LTD lobe, the STDP protocol using the SF seems to have a vertical shift, in which long pre-before-post or post-before-pre delays result in a reduction in the synaptic weight, unlike what I've seen in standard STDP protocols, where long delays do not result in a weight change. Can the authors explain this discrepancy between standard STDP results and those produced by the SF?

One of my major issues with the paper was that I don't think the authors have remotely established neural plausibility. First, the dimensionalities used, <20, seem to be significantly lower than those relevant to most tasks real neural systems would be solving, even accounting for sparsity. Perhaps the authors can be more specific about the setting in which they think their rule would apply? The authors suggest some ways to address this at the end of section 2.2, but I think, given the importance of these issues to biological plausibility, the suggestions should be fleshed out and incorporated into the paper. Also, the performance of the model in cases of model mismatch seems to drop rapidly with dimensionality, as shown in Figure 2E. Since it's likely that the brain is operating under model mismatch, it is alarming to see the performance of the SF drop so precipitously with the level of mismatch. The authors counter that the performance is relative to that of the gradient-based rule with optimized learning rates, but it would seem that optimizing a single learning rate can be done much more plausibly than tracking a full covariance matrix. This brings me to my second issue regarding neural plausibility: the SF is indeed normative, but requires the tracking of covariances between synaptic weights. How do the authors imagine this could possibly done in a realistic biological setting? The authors are aware of this problem and mention some possible solutions in the discussion, but I would want these to be pursued in the main text, because without some hope of neural plausibility, I don't see how the SF, despite its normative benefits, is neuroscientifically relevant. This problem is particularly acute because the authors, to their credit, have investigated the performance of the diagonal synaptic filter, which would be more plausible. Unfortunately, as the authors show, its performance is often significantly worse than that of the SF, and also cannot explain heterosynaptic plasticity. Thus it seems that tracking of non-diagonal covariances is crucial, but it seems entirely unclear how it would be done in a biologically plausible setting.

Given the biological plausibility issues above, I also felt that the paper did not adequately test the SF against other, non-normative, models that _are_ biologically plausible. For example, I believe the LTD lobe of STDP can be produced within the author's own framework by holding the variance constant at the identity and just updating the means. The authors themselves cite existing work in the literature outside of the normative that can also reproduce this lobe. Holding the variance constant is a (limited) example of the diagonal synaptic filter, which the authors show cannot reproduce heterosynaptic plasticity. But perhaps this latter effect can be explained by applying sparsity inducing priors to the weights; intuitively it would seem that such a prior would also induce heterosynaptic plasticity, but have a more plausible biological implementation. Of course, the advantage of the SF is not just that it reproduces certain observed effects, but it does so in a principled way. Nevertheless, given how hard it seems to be to implement the normative solution in a biologically plausible way, I think it's important to rigorously test the SF against non-normative alternatives, as these alternatives might reflect the biologically plausible solutions that the brain has developed to solve the problems the SF addresses, but in realistic settings.

Thus in summary, I like (1) the normative framework the SF provides, (2) its explanation for heterosynaptic plasticity, (3) that it, at least partially, captures the negative lobe of the STDP curve, and (4) that it makes testable predictions about weight change variance in STDP experiments, and I think the authors have the core of a strong manuscript here. However, I think they need to do significantly more work to establish the advantages and limitations of the SF in neurally plausible settings, such as rigorous testing of their SF in more realistic settings with larger dimensionality and with more and different types of model mismatch, and a broader comparison to the best available non-normative competitors. Only then will I be able to accept that the Synaptic Filter provides, in the words of the authors, "a serious candidate for a computational principle underlying plasticity."

On a non-scientific note, I also want to mention that I found the presentation of the paper to be quite poor. Because the technical content of the paper was correct (I manually checked most of the equations) I did not let the poor presentation of the paper affect my decision. Nevertheless, the problems below suggest a lack of both attention to detail and respect for the reviewers' time and need to be addressed before any resubmission.

The figures: These were provided unlabeled and appended in low resolution to the end of the paper as figS3, figS5, figS4, figS1, figS2, fig3, fig2, fig1, fig4. The proper labels could only be determined from the hyperlinks to the corresponding high-res sources provided in the corner of each figure. Next, the figures were provided in the sequence above, i.e. out of order. Further, the figures were mislabelled: figures S1-S5 are referred to as figures 5-9 in the main text. I'm not sure how this happened, whether on the journal's end or the authors, but it has not happened on previous papers I have reviewed for this journal. I had to spend a significant amount of time before I could even start the review to collect and assemble the figures in the correct order. Providing figures in this format suggests a lack of respect for my time as a reviewer. In the future, please ensure that high-quality vector graphics figures with the proper labels are provided in the correct order.

The text: First, the citation format used in the paper is bulky and inconsistent, frequently appending the last names of all authors to the citation. For example, on line 166, the paper by Liakoni et al. 2019 is referred to as "Liakoni et al. 2019 Liakoni, Modirschanechi, Gerstner, Brea". This is obviously not the desired format for the citation (why list Liakoni twice, why use 'et al.' but then list all the authors anyway), and the error is obvious with even a cursory glance at the paper. Secondly, the paper has numerous minor errors. Most, but not all, were errors in spelling or grammar. In most cases these did not detract from the understandability of the content, but did from its readability. I've listed the ones I could find at the end of this review. Some of these errors weren't even buried in the text, but in section headings. Please ensure that the manuscript is properly proof-read before any future resubmission.

I thank the authors again for their submission.

----

List of minor errors found:

- Abstract: "a better generalization performance" should be "better generalization performance"

- Abstract: "consistent with the STDP" should be "consistent with STDP"

- 106: No comma after "generative model"

- 199: For all four filtering models .... as depicted in Figure 2B: Only two models are shown in that panel, not four.

- 237: "in presence" should be "in the presence"

- 261: Synpatic should be Synaptic.

- 275: "differences in model evidence can be more directly attributed". More directly than what? Presumably MSE? But the sentence before just said that MSE is not even defined when there's model mismatch. So get rid of 'more' and just say 'can be directly attributed'

- 283: "less dimensions" should be "fewer dimensions"

- 304: Missing 'with': Should be "The Synaptic Filter is consistent with STDP"

- 319: "Matter debate" should be "Matter of debate"

- 338: "experiment" should be "experiments".

- 354: "is consistent" should be "are consistent."

- 373: "inputs spikes" should be "input spikes"

- 467: "induction mechanism. In the case" should be "induction mechanism, in the case"

- 473: "stimulated" should be "simulated"

- 495: "Parametrising" should be "Parameterising"

- In section 4.1.2, on the second line D_t := {x_t, y_t} : The t subscripts on x and y should be taus.

- Equation 8 in that section: Some mention should be made that the proportionality there is only approximate, because x_t^eps is being used not x_t.

- Two lines after equation 8: Bernulli should be 'Bernoulli'

- 661: The identity matrix is typically blackboard I, not 1.

- 4.3.1, equation 16: The equality should be replaced by approximate equality, because it holds when tau_ou >> tau_m. Similarly in equation 17. Alternatively, the authors can just state somewhere that they're going to use equality to mean approximate equality in these cases.

- Equation 17: The quantity is the variance of u_t, not u_t^2.

- Further down, fourth line after equation 19, the variance is that of u_t, not u_t^2.

- Equation 20: Given Equation 19, the expression for c in Equation 20 is missing a beta_0 in the denominator.

- Equation 21: The statement before the equation is "Assuming a flat model prior, we have", but then what's written in the equation is the log _likelihood_ of the model, which is what it is regardless of the prior. I think what's mean is that in the case of a flat model prior, the log model evidence is equal to the log likelihood up to constant term. So perhaps the equation should be written in terms of the log evidence as log p(M|D_t) = ... + constant.

- 1064: "and it cannot influence" should be "and cannot influence"

- 1130: asymtotically should be asymptotically

- 1133: "higher dimension" should be "higher dimensions"

- Caption for Fig 6 near line 1151: "all filtering distribution" should be "all filtering distributions"; "particle filer" should be "particle filter"

- Line after eqn. 30: "dependent" should be "depend"

- Eqn 44: The second equation is for dSigma_t, not dSigma_t^2

- Line immediately before Eqn. 57: dtdelta should be ddelta.

- Eqn 57: This equation is unclear; first, what does the proportionality mean? Eqn 44 is exact, and all that's happening here is a substitution. Secondly, the terms proportional to dt in Eqn 44 are missing here; they reappear in Eqn. 59. If the intention is to discuss only the terms proportional to ddelta nd dN here, then a different symbol should be used. It's probably easiest and clearest to just write down what dSigma is after making all the substitutions, rather than leave parts out.

- 1244: "predictive of the gradient" should be "predictive performance of the gradient"

- 1254: "selected its maximum value of the loglikelihood" should say "selected the maximum value of the fit to the loglikelihoods."

- 1260: "to maximum value" should be "to the maximum value"

- 1280: "negative lobe dependence" should be "negative lobe depends"

- 1281: "do not" should be "does not"

- Caption to Fig. 8 (following line 1295): "The returns to its" should be "The mean value of the bias returns"

- The authors mention Kushner 1964 as an early example of filtering in the literature, but what about Norbert Wiener's 1949 "Extrapolation, Interpolation, and Smoothing of Stationary Time Series"?

**Have all data underlying the figures and results presented in the manuscript been provided?**

Reviewer #1: None

Reviewer #2: Yes

Reviewer #3: None

PLOS authors have the option to publish the peer review history of their article (what does this mean?). If published, this will include your full peer review and any attached files.

Reviewer #1: No

Reviewer #2: No

Reviewer #3: **Yes: **Sina Tootoonian
---

## [Decision Letter · Decision Letter 1]

1 Oct 2021

Dear Dr Jegminat,

Thank you very much for submitting your manuscript "Learning as filtering: implications for spike-based plasticity" for consideration at PLOS Computational Biology. As with all papers reviewed by the journal, your manuscript was reviewed by members of the editorial board and by several independent reviewers. The reviewers appreciated the attention to an important topic. Based on the reviews, we are likely to accept this manuscript for publication, providing that you modify the manuscript according to the review recommendations.

In particular, you need to address Reviewer 2's argument that the claim that SF outperforms GD is unsupported, and Reviewer 3's points regarding biological plausibility. In these cases, you can probably address these comments in the text by, e.g., toning down or taking out claims of outperforming GD and discussing candidly potential biologically implausible components in the model.

Sincerely,

Blake A Richards

Associate Editor

PLOS Computational Biology

Samuel Gershman

Deputy Editor

PLOS Computational Biology

[LINK]

Reviewer's Responses to Questions

**Comments to the Authors:**

Reviewer #1: I am satisfied with the adapted changes.

Reviewer #2: Review is uploaded as an attachment.

Reviewer #3: I thank the authors for their submission of the revised manuscript, (and for including the fully compiled manuscript, which made reading the submission much easier). I appreciated the focus on characterization using MSE, and the extended characterization of performance with respect to dimension. As in the original submission, I also very much appreciate the normative approach to synaptic plasticity proposed here. However, as I'm sure the authors will agree, the normative approach is most illuminating when the resulting derivation is biologically plausible. I don't think the authors have established plausibility, as I describe below.

First, the strengths: I believe these are:

- The normative derivation, and the 'learning as filtering' approach.

- The explanation of the negative lobe of STDP;

- The explanation of heterosynaptic plasticity

- The prediction about spike-timing-dependent changes in EPSP variance.

- The elegant derivation of the Block Synaptic Filter by projecting the covariance dynamics of the full synaptic filter into the space of symmetric, block-diagonal matrices; also the potential biological implementation of this filter by synapses on dendritic branches.

- The discovery of one regime of intermediate signal dimensionality where the block and full synaptic filters clearly outperform the diagonal synaptic filter (Fig. 2E).

The main weakness of the model remains its biological plausibility. The original model (the Full Synaptic Filter) was implausible for two reasons: (1) it requires the storage of O(d^2) covariances for d synapses, and (2) these values, as covariances, are specific to pairs of synapses. The authors have addressed the first problem by deriving the Block Synaptic Filter, which requires fewer values to be stored, a number which importantly is now linear in d instead of quadratic. However, they have not addressed (nor even acknowledged) the second problem of how it would be possible not only for pairs of synapses, even those on the same dendritic branch, to store and update pair-specific covariance values. This might be possible if the synapses are immediately adjacent -- corresponding to a tridiagonal structure of the covariance matrix -- but otherwise it's difficult for me to see how non-adjacent synapses on a branch could coordinate each of their pairwise covariances independently. I want to emphasize that I'm not asking for biophysical plausibility (nor was I in my original review) -- but the less strict 'biological plausibility': whether it is at all possible for synapses on the same branch that are not immediately adjacent to share and update pair-specific quantities.

Because synaptic machinery is complex and is still being explored, the apparent implausibility may perhaps be rendered plausible by some as yet unknown (at least to me) inter-synaptic mechanism. Therefore, I would ask that the authors at least flesh out in some detail what the biological requirements for their update rules are, in particular the requirement implied by Eqn 6 for some form of communication between synapses to store the covariances. Additionally, equation 5 seems to require that even the mean value of each synapse is a linear weighting of the input at all synapses x_t by the appropriate elements of the synaptic covariance matrix. This again would seem very difficult to implement given what's known (at least to me) about synaptic and inter-synaptic machinery. Again, I'm not asking for full biophysical details, but just spelling out who needs to communicate what with whom.

On a lesser note, the authors show only one regime where the block and full synaptic filters significantly outperform the (much more plausible) diagonal synaptic filter (Figure 2E). This was in the setting in which the input had the block-diagonal covariance structure suitable for the Block Synaptic Filter. Even though the covariance in the model is meant to capture both environemental dynamics and parameter uncertainty, I wondered why the authors didn't also try a setting in which the weight dynamics (Eqn 1) included correlations? This might have allowed the three models to be better distinguished.

In summary I appreciate the normative derivation and the additional exploration of the performance of the Synaptic Filter but it still remains biologically implausible given current knowledge (at least mine) of synaptic and dendritic mechanisms, so I ask that the authors at least specify in some detail what the biological requirements for their model are.

Minor points:

- The authors should explain better their differential-geometric derivation of the block-synaptic filter. For example, why is the procedure described on lines 1107-1110 the right thing to do? My guess is that the tangent space to the manifold of symmetric positive-definite matrices (i.e. covariance matrices) is isomorphic to the space of symmetric matrices Sym(d), and we want to project elements of this tangent space onto a block-diagonal subspace. Why is the Fisher-Rao metric the right metric with which to do this?

- On line 224 the authors claim a particular difference in performance is 'significant'. Do they mean statistically significant? If so, please provide the statistics. If not, please clarify.

- On line 233 the authors say: "contrary to the assumption of homogeneous, constant input rates ρ, neuronal activity in the brain is often sparse, i.e., most neurons are silent most of the time." But there is no contradiction between 'Most neurons being silent most of the time' 'homogeneous, constant input rates', only requires that these rates be low.

- Line 204,252: 'block' should be 'Block'

- Line 237: 'dependent' should be 'depend'

- Line 368: 'competes' should be 'computes'

- Line 501: 'don't' should be 'doesn't'

- Line 611: 'burnin' should be 'burn-in'

- Line 858: First quotation mark is backwards

- Line 925: 'burn in' should be 'burn-in'

- Line 1114: There's an extra tab before 'where'

**Have the authors made all data and (if applicable) computational code underlying the findings in their manuscript fully available?**

Reviewer #1: Yes

Reviewer #2: Yes

Reviewer #3: None

PLOS authors have the option to publish the peer review history of their article (what does this mean?). If published, this will include your full peer review and any attached files.

Reviewer #1: No

Reviewer #2: No

Reviewer #3: **Yes: **Sina Tootoonian

Figure Files:

Data Requirements:

Reproducibility:

References:

---

## [Editor Report · Decision Letter 2]

3 Dec 2021

Dear Dr Jegminat,

We are pleased to inform you that your manuscript 'Learning as filtering: implications for spike-based plasticity' has been provisionally accepted for publication in PLOS Computational Biology.

Best regards,

Blake A Richards

Associate Editor

PLOS Computational Biology

Samuel Gershman

Deputy Editor

PLOS Computational Biology

---

## [Editor Report · Acceptance letter]

14 Jan 2022

PCOMPBIOL-D-20-01490R2 

Learning as filtering: implications for spike-based plasticity

Dear Dr Jegminat,

I am pleased to inform you that your manuscript has been formally accepted for publication in PLOS Computational Biology. Your manuscript is now with our production department and you will be notified of the publication date in due course.

With kind regards,

Zsofia Freund
